# *Listeria monocytogenes* Biofilms Are Planktonic Cell Factories despite Peracetic Acid Exposure under Continuous Flow Conditions

**DOI:** 10.3390/antibiotics12020209

**Published:** 2023-01-19

**Authors:** Kyle B. Klopper, Elanna Bester, Gideon M. Wolfaardt

**Affiliations:** 1Department of Microbiology, Stellenbosch University, Stellenbosch 7600, South Africa; 2Department of Chemistry and Biology, Toronto Metropolitan University, Toronto, ON M5B 2K3, Canada

**Keywords:** *L. monocytogenes* EGD-e, *Listeria monocytogenes* biofilms, peracetic acid, food processing environment, biofilm metabolic activity, biofilm biomass, planktonic cell yields, disinfection, 10 °C, 37 °C

## Abstract

*Listeria monocytogenes* biofilms are ubiquitous in the food-processing environment, where they frequently show resistance against treatment with disinfectants such as peracetic acid (PAA) due to sub-lethal damage resulting in biofilm persistence or the formation of secondary biofilms. *L. monocytogenes* serovar ½a EGD-e biofilms were cultivated under continuous flow conditions at 10 °C, 22 °C, and 37 °C and exposed to industrially relevant PAA concentrations. The effect of PAA on biofilm metabolic activity and biomass was monitored in real-time using the CEMS-BioSpec system, in addition to daily measurement of biofilm-derived planktonic cell production. Biofilm-derived planktonic cell yields proved to be consistent with high yields during biofilm establishment (≥10^6^ CFU.mL^−1^). The exposure of biofilms to the minimum inhibitory PAA concentration (0.16%) resulted in only a brief disruption in whole-biofilm metabolic activity and biofilm biomass accumulation. The recovered biofilm accumulated more biomass and greater activity, but cell yields remained similar. Increasing concentrations of PAA (0.50%, 1.5%, and 4.0%) had a longer-lasting inhibitory effect. Only the maximum dose resulted in a lasting inhibition of biofilm activity and biomass–a factor that needs due consideration in view of dilution in industrial settings. Better disinfection monitoring tools and protocols are required to adequately address the problem of *Listeria* biofilms in the food-processing environment, and more emphasis should be placed on biofilms serving as a “factory” for cell proliferation rather than only a survival mechanism.

## 1. Introduction

*Listeria monocytogenes* is a Gram-positive opportunistic pathogen and the causative agent of listeriosis in humans and some animals. As such, it is often associated with contaminated food or food production facilities, especially ready-to-eat foods [1,2,3]. *L. monocytogenes* is well-adapted to various environments (meat, vegetables, soil, water, etc.) with varying physical attributes (a wide range of temperatures, salt concentrations, pH, etc.), which renders it particularly capable of colonizing and persisting in diverse settings [1,2,3,4]. The intrinsic ability of *L. monocytogenes* to grow unimpeded under refrigeration conditions (0–10 °C) contributes to its persistence and recalcitrance in the food processing environment [5,6,7]. This, coupled with *Listeria’s* temperature-dependent motility (motile ≤ 30 °C and loss of motility ≥30 °C) and increased virulence at mammalian body temperature (37 °C), results in a greater risk of contracting listeriosis from the consumption of contaminated or inadequately cooked foods [5,6,7,8].

The formation of sessile communities by *L. monocytogenes* on various surfaces further contributes to the ubiquity of this organism [1,9,10]. Biofilms are complex structures of highly variable intra- and inter-species aggregates of cells that are encased in equally complex and variable extracellular polymeric substances/matrices (EPS) [11,12,13,14,15]. Bacteria preferentially exist and proliferate as complex community structures, such as flocs and biofilms, rather than the scientific normal of planktonic growth in batch mode [12,13,14]. The multifaceted and diverse nature of sessile communities contributes to their recalcitrance and allows for the colonization of diverse environments [5,6,9,14,15,16]. These complex sessile communities prove to be a substantial obstacle to adequate and sustained disinfection of the food processing environment (FPE) despite the availability of various disinfectants and dispersants [3,5,6,7,9,10,15].

Recalcitrant *L. monocytogenes* biofilms are ubiquitous in FPEs, with reports of their persistence for months or even years in these settings [6,10]. *L. monocytogenes* is particularly persistent at refrigeration temperatures, low pH, heat, and desiccation, and its persistence is further enhanced by its ability to form biofilms [6]. Sessile communities of *Listeria* spp. typically show a greater tolerance to sanitizing agents (quaternary ammonium compounds, peracetic acid, etc.) than their planktonic counterparts [10].

Peracetic acid (PAA) is a broad-spectrum, strong oxidizing class of peroxy acid that is frequently used as a disinfectant of hard surfaces in FPEs and as surface disinfection of fresh produce [1]. This broad spectrum of activity can be attributed to its functionality across a wide temperature range (0 to 40 °C) and the oxidation of cellular components, including proteins and nucleic acids [1,2,3]. PAA is known as an effective sanitizer for use against *L. monocytogenes,* especially in the food industry. However, it has been noted that PAA can induce sub-lethal injury in pathogenic organisms [1,2,4]. This, coupled with the severe reduction in PAA oxidation capacity in the presence of organic substances, diminishes the disinfection efficacy of PAA in environments with high organic contents, such as that found in biofilm biomass [4].

Operational and abiotic factors such as incorrect/poorly timed dosage, inadequate rinsing, as well as physical and chemical properties of surfaces often result in the application of sub-lethal concentrations of disinfectants [6,10]. Insufficient or sub-lethal disinfection results in only partial killing of *Listeria* biofilms, which may cause the persistence of dead cells/cell debris in the biofilm matrix, an increase of surviving viable cells, and the creation of a microhabitat for subsequent re-colonization [6,10].

The ubiquitous nature of biofilms can be viewed through two distinctive mechanistic approaches. The conventional approach views biofilms primarily as a survival mechanism, notably due to their increased tolerance or resistance to antimicrobials [9,10,11]. A more contemporary approach views biofilms as a mechanism for microbial proliferation due to planktonic cell production and release to the bulk fluid [11,17,18]. Biofilms can shed significant numbers of cells, typically in excess of 10^6^ CFU.mL^−1^ under continuous flow conditions [17,18]. The potential for these cells to spread in a FPE and initiate biofilm formation on new surfaces should be a pressing concern.

Biofilm EPS comprises the greatest proportion (≥90%) of total biofilm biomass and aids in the duality of biofilms as a means of survival and proliferation [4]. EPS acts as a complex housing structure providing protection from the detrimental effects of various antimicrobial agents [4,9,10], yet without acting as a barrier that prevents continuous cell release from the biofilm [11,15,17,18]. Although antimicrobial agents may inhibit/kill viable cells in the biofilm, they often fail to affect the EPS component of the biofilm biomass. The ineffective removal/reduction in EPS contributes substantially to a biofilm’s recalcitrance.

The ineffective removal of the biofilm biomass and death/inhibition of only a proportion of viable cells (metabolic dormancy) when exposed to sub-lethal concentrations of antimicrobials or antimicrobials without dispersant properties can result in the phenomenon of “secondary biofilms” [4]. These secondary or tenement biofilms provide an existing EPS structure (albeit somewhat damaged) for metabolically dormant, sub-lethally damaged cells to recover or new cells to colonize, thereby resulting in biofilm recovery in addition to contributing to the proliferation of planktonic cells. The purpose of this study was, therefore, to elucidate mechanisms contributing to the recalcitrant nature of *L. monocytogenes* biofilms in the FPE. Two potential causes for persistence were considered, namely (i) that biofilms serve as highly productive ‘cell factories’ releasing actively growing cells and (ii) that exposure of these biofilms to sub-lethal concentrations of food-grade disinfectants such as PAA leads to the formation of a transient, surface-associated matrix that harbors viable and damaged cells with the capacity to rapidly respond to favorable growth conditions.

## 2. Results

In order to delineate the dual survival-proliferation function of biofilms, their biomass and metabolic activity were quantified in real-time under continuous-flow conditions. The response of the in-situ biofilm biomass and metabolic activity to various concentrations of the PAA-based sanitizer was measured using the combined CEMS-BioSpec system previously developed in our group [19,20,21]. Briefly, the system consists of two carbon dioxide evolution measurement systems (CEMS), each comprised of a silicone tube as a continuous-flow bioreactor encased in an outer Tygon tube, which facilitates the removal of microbially-derived CO_2_ by a sweeper gas and subsequent analysis by a CO_2_ analyzer (Figure 1) [19,21]. This system is premised on the high CO_2_ permeability of silicone (permeability coefficient of 20132) vs. near-impermeability of Tygon tubing (permeability coefficient of 270). The BioSpec component of the CEMS-BioSpec system is based on the well-established use of light absorbance/transmission/scattering to measure microbial biomass [20]. Briefly, BioSpec was inserted in-between the two CEMS systems and consisted of silicone tubing sandwiched between a perpendicularly oriented amber LED (595 nm) on one side of the tube and a high-accuracy digital light sensor on the opposite side of the tube (Figure 1).

### 2.1. Minimum Inhibitory Concentration (MIC) Determination

The results based on the microbroth dilution method showed that temperature had a relatively minor effect on the MIC of PAA on *L. monocytogenes* serovar ½a EGD-e at the three temperatures tested (Figure 2). However, the effect at sub-MIC levels was varied. For instance, while significant inhibition of growth at the coldest temperature (10 °C) was evident at a PAA concentration of 0.02%, the MIC was 3-fold higher (0.16% PAA) (Figure 2a). Incubation at room temperature (22 °C) resulted in a doubling of the lowest concentration where growth was affected, to 0.04% PAA (Figure 2b). However, the MIC was the same as that of the refrigeration temperature. Cultivation at body temperature (37 °C) had a substantial effect on growth inhibition (Figure 2c), with a reduction of the MIC to 0.08% PAA (2× lower than that observed at 4 °C and 22 °C, Figure 2a,b). Overall, it was decided to select 0.16% PAA as the MIC since it was effective at inhibiting growth across the temperature range applied in this study.

### 2.2. Effect of PAA on Cell Release, In-Situ Biofilm Biomass and Metabolic Activity under Continuous-Flow Conditions at Different Temperatures

Biofilms cultivated under continuous flow conditions approached a steady state with respect to metabolic activity and biofilm biomass between approximately 50 and 120 h of cultivation at the three temperatures at which the experiments were conducted (Figure 3, Figure 4 and Figure 5). In contrast, the planktonic cell numbers in the effluent typically leveled off at ~10^8^ cells per ml after 48 h and remained remarkably constant thereafter. Since the displacement rate of the growth medium through the CEMS-BioSpec system was double the maximum growth rate of *L. monocytogenes*, it can be assumed that the planktonic cells in the effluent were primarily produced and released by the biofilms. The respective durations to approach steady state were used as a guide for when to apply the single treatment with PAA in order to simulate the treatment of established biofilms in practice.

#### 2.2.1. Biofilms Cultivated at 10 °C

*Exposure of the biofilms to the MIC concentration of PAA (0.16%).* Treatment resulted in an immediate ~1-log increase in planktonic cell yield as counted after 1 h (displacement of three reactor volumes; t = 121 h. Figure 3a), probably due to the release of weakly-bound cells by PAA, which is a strong oxidant. This was followed by a 2-log reduction in planktonic cell numbers after a further 23 h (t = 144 h), where the numbers returned to pre-treatment values (Figure 3a). Treatment had a pronounced inhibitory effect on the metabolic activity of the biofilm, which was accompanied by a reduction in biofilm biomass over the first 12 h after dosing, resembling sloughing events as indicated by the black and red lines in Figure 3b, respectively. However, the activity rapidly recovered, and surface-attached biofilm biomass increased over the 24 h after dosing, leading to the formation of more biofilm biomass than that of the pre-dose steady state (from 145 h, red line, Figure 3b). Of note is the relative stability in planktonic cell yield after recovery, compared to the pronounced inflection points measured for metabolic activity and biofilm biomass. Earlier work also reported this phenomenon; for instance, Kroukamp et al. [22] described similar sloughing events of which the severity increased with increasing nutrient concentrations.

*Exposure of biofilms to 0.5% PAA.* This is the minimum dose recommended by the manufacturer. Similar to the treatment with 0.16% PAA, the 0.5% PAA dose also resulted in a temporary increase in planktonic cell yields (121 h, Figure 3c), followed by a more pronounced, 3-log reduction in numbers at 144 h. Again, the reduction was followed by prompt recovery to pre-dose cell yields (±10^8^ CFU.mL^−1^). Although there was a brief increase in biofilm biomass (121 h, red line), the treatment had an inhibitory effect on both biofilm biomass and metabolic activity, with the respective metabolic inhibitory and biofilm removal effect lasting for ±20 and 24 h (Figure 3d).

*Exposure of biofilms to 1.5% PAA (10-fold increase of the MIC).* Planktonic cell yield from the sessile population (Figure 3e) followed the same pre-exposure trends as seen for the two lower doses of PAA. While the cell numbers derived from the biofilms were unchanged an hour after exposure, there was a complete loss in cell yield for the following 48 h (144 to 192 h, Figure 3e), which coincided with the inhibition of metabolic activity and biomass (Figure 3f). The cessation in planktonic cell production and release from the biofilms was temporary since 10^6^ CFU.mL^−1^ was present in the system effluent 72 h after dosing, with complete recovery of planktonic cell production to pre-dosing levels within 96 h (216 to 360 h, Figure 3e). The decrease in metabolic activity to baseline-detection levels (Figure 3f, black line) lasted longer than that observed with the MIC dose (Figure 3b, black line); ±80 h after 1.5% PAA vs. ±5 h after 0.16% PAA. The higher PAA concentration also resulted in a decrease in biofilm biomass during dosing, which was similarly followed by a period of biofilm accumulation inhibition for ±80 h (red line, Figure 3f). The recovery in biofilm metabolic activity at ±200 h slightly preceded that of the biofilm biomass, which continued to increase over the duration of the recovery period. Both biofilm metabolic activity and biomass reached higher steady-state levels after recovery from exposure to the increased PAA concentration than pre-exposure levels.

*Exposure of biofilms to 4% PAA (maximum recommended application dose).* As shown in Figure 3g, planktonic cell yield was reduced from ± 10^9^ CFU.mL^−1^ to zero following dosing (Figure 3h). This loss of cell production was maintained for 168 h before it increased to 10^5^ CFU.mL^−1^. An unexpected result, though, was the fact that biofilm biomass did not show a rapid decrease as measured following the lower dosages but instead gradually decreased to baseline values 300 h after dosing and then increased. However, this increase was not preceded or accompanied by an increase in metabolic activity. After activity rapidly dropped, as seen for the lower PAA concentrations, it never recovered (black line, Figure 3h). In contrast to the sustained inhibition of metabolic activity, an increase in absorbance was observed from 300 h onwards, in sync with the observed increase in planktonic cell numbers (red line, Figure 3h,g).

#### 2.2.2. Comparison to Biofilms Cultivated at Higher Temperatures

Being a psychrotolerant bacterium that can grow at temperatures around 0 °C, *L. monocytogenes* poses serious challenges to the food industry, especially when refrigeration systems underperform. Its adaptive response to cold is well described, including decreased metabolic rates [23]. We, therefore, selected 10 °C for this study, which is slightly above the typical refrigeration range used in industry. The description in the previous section of biofilm response following exposure to four PAA concentrations at 10 °C is used here as the basis for comparison to higher temperatures, with the focus switching to trends among the three temperatures. Monitoring, as reported in Figure 3, continued for up to 240 h after treatment to assess the potential for subsequent damage that could present itself later. Because the biofilms approached a steady state earlier at the higher temperatures, PAA treatment was introduced earlier (at 72 h, 96 h, and 120 h for experimentation at 37 °C, 22 °C, and 10 °C, respectively), and therefore the experiments at higher temperatures were terminated earlier. The time of termination of each experimental run was set on 4 days post recovery of planktonic cell yield following PAA treatment.

The pre-exposure trends in planktonic cell production by the biofilms cultivated at higher temperatures showed a similar trend to that of 10 °C, with an average yield between 10^8^ and 10^9^ CFU.mL^−1^ (Figure 4a,c,e,g and Figure 5a,c,e,g).

*Exposure of the biofilms to the MIC concentration of PAA (0.16%).* Planktonic cell production at 22 °C did not change after PAA exposure (Figure 4a, 120 h), whereas that of the 37 °C biofilm decreased by 1 log 24 h after exposure (Figure 5a, 96h). The lower level of production was maintained for 48 h prior to recovery to pre-exposure levels of 10^9^ CFU.mL^−1^ at 144 h (Figure 5a). The higher temperature of 22 °C and 37 °C resulted in established biofilms 24 to 48 h earlier than at 10 °C (96 h and 72 h, respectively, Figure 4b and Figure 5b). The exposure of the sessile community to the MIC concentration of PAA at 22 °C had a similar effect to that seen at the lower temperature; a prompt but short-lived decrease in metabolic activity followed by recovery in activity. In contrast, when biofilms cultivated at 37 °C were exposed to this PAA concentration, it caused a prompt decrease in metabolic activity (±72 to 75 h) with a reduction to the pre-establishment levels, which lasted for 24 h followed by recovery (black line, Figure 5a). The biofilms all recovered to either pre-exposure or higher metabolic levels during the recovery phase, as also seen in the refrigerated temperature biofilms (black lines, Figure 4b and Figure 5b). The introduction of the PAA at 96 h and 120 h had minimal effect on the biomass, with only a slight peak and trough observed. However, at 37 °C, the biomass decreased over a 24 h period (red line, Figure 5b). This suppression of biomass was temporal, with an increase observed from ± 90 h (18 h after dosing). Overall, it is plausible that the rapid recovery in activity was facilitated by the remaining biomass, along with an increase in the amount of biomass itself to a level substantially higher than before exposure (red lines, Figure 3b, Figure 4b and Figure 5b).

*Exposure of biofilms to 0.5% PAA (minimum manufacturer-recommended dose).* Similar to the treatment of the 10 °C biofilm with 0.5% PAA, the exposure of the 37 °C biofilms to the same concentration of PAA also resulted in a temporary increase in planktonic cell yield (Figure 5c, 73 h). Planktonic cell production decreased 24 h after dosage, but to a greater extent in the case of the 37 °C biofilm (3 log reduction, Figure 5c, 96 h) than that of the 22 °C biofilms (1 log reduction, Figure 4c, 120 h). Yield at all three temperatures recovered to pre-dosage concentrations 48 h after exposure. Whole-biofilm metabolic activity at 37 °C exhibited a similar trend to that at 22 °C, with a prompt reduction in the rate of CO_2_ produced by the biofilms to baseline (Black lines, Figure 4b and Figure 5b). The suppression of metabolic activity at 22 °C was relatively short (≈16 h), while that of 37 °C was nearly twice as long (≈30 h). The metabolic activity recovered promptly, albeit to a new steady state that was slightly lower than that observed pre-dosing (black line, Figure 4b and Figure 5b). The changes in biofilm biomass mirrored the metabolic activity well with a prompt and sustained decrease in absorbance post-exposure to the higher concentration of PAA (red line, Figure 4b and Figure 5b). After this biocide-induced suppression, the biofilm biomass promptly recovered to a greater extent than before exposure (red line, Figure 4b and Figure 5b). Overall, the metabolic activity and biofilm biomass responded with a high degree of similarity irrespective of the temperature and recovered substantially better than at the refrigerated temperature.

*Exposure of biofilms to 1.5% PAA (10-fold increase of the measured MIC).* The effect of this concentration of PAA on planktonic cell yield was similar for biofilms cultivated at 10 °C and 37 °C. Production was inhibited for 48 h post-dosing (Figure 3e, 144 to 168 h and Figure 5e, 96 to 120 h) prior to its recovery to pre-dosage levels. The higher PAA concentration did have a greater effect on production from 22 °C biofilms than that of the lower concentrations, with a 4 log reduction 24 h after exposure (Figure 4e, 120 h), but this effect only lasted for 24 h. A similar response in metabolic activity was noted, as at the lower PAA concentration (0.5%), with a prompt decrease in CO_2_ production rates as well as the biofilm biomass (Figure 4f and Figure 5f). Interestingly, while the suppression of metabolic activity at 37 °C lasted for a similar period as that of the same concentration applied at 10 °C (≈72 h), this period was substantially shorter (≈36 h) for the room temperature biofilms (black line, Figure 4c). During the recovery phase, only the biofilms cultivated at 10 °C and 22 °C recovered to pre-dose levels (Figure 3f and Figure 4f), whereas the 37 °C biofilms did not recover during the experimental period (≈half of the pre-dose level, Figure 5f). A decrease in biofilm biomass due to the oxidative action of the PAA was evident after the dosing period, followed by a period of oscillation during the recovery phase (red lines, Figure 4f and Figure 5f). The biomass of biofilms cultivated at 22 °C only appeared to recover fully toward the end of the experimental run. In contrast, the amount of attached biomass at 10 °C and 37 °C oscillated after treatment, but these sloughing events never resulted in a complete loss of biomass (red line, Figure 5f).

*Exposure of biofilms to 4% PAA (maximum recommended application dose).* The highest PAA concentration had the greatest effect on planktonic cell production, with the yield decreasing to zero for a 48 h-period post-dosing for both the 22 °C and 37 °C biofilms (Figure 4g, 120 to 144 h and Figure 5g, 96 to 120 h). The inhibitory effect did not, however, last as long as that of the 10 °C biofilm (Figure 3g, 144 to 264 h). Dosing promptly reduced metabolic activity to baseline for both the 22 °C and 37 °C biofilms (black lines, Figure 4h and Figure 5h). Metabolic inhibition was maintained for ~ 64 h prior to recovery for the 22 °C biofilms (Figure 4h). In contrast, metabolic activity did not recover during the experimental period for the 37 °C biofilm (≥160 h, Figure 5h). Biofilm biomass initially decreased for both the 22 °C and 37 °C (red line, Figure 4h and Figure 5h). The biofilms cultivated at 37 °C started to recover within 24 h (red line, Figure 5h) and that of the 22 °C within 48 h of dosing (red line, Figure 4h). Recovery continued despite pronounced sloughing events and exceeded pre-dosage levels. The oscillation in biomass values is possibly due to the ‘tumbleweed’ phenomenon. These *Listeria* biofilms had a distinct feature reminiscent of a tumbleweed plant in that they accumulated as clusters, visible to the unaided eye, that periodically moved downstream (Figure 6).

As shown in the control experiments (Appendix A), dH_2_O, the diluent used to constitute the working stocks of various concentrations of PAA, had minimal effect on biofilm activity, biofilm biomass, and biofilm-derived planktonic cell yield. Similarly, PAA had a negligible effect on these parameters when introduced into a sterile system and therefore had no influence on the results related to biotic response.

## 3. Discussion

In general, the duration of metabolic inhibition post-treatment increased with increasing concentrations of PAA at all three cultivation temperatures. The same can be said for planktonic cell production, with some exceptions (e.g., Figure 4e). While the amount of attached biomass followed similar trends to that of activity, the response was less predictable and more variable. This may be due to the ‘tumbleweed’ phenomenon, where the biomass did not cover the surface in a uniform manner but rather in distinct clumps that grew in size and periodically detached and flowed downstream. As shown in Figure 4 and Figure 5, the biofilm-to-effluent cell yield at 22 °C and 37 °C was remarkably similar to those at 10 °C. Previous studies e.g., [11,18,20] reported similar levels of cell yield and the role thereof in proliferation by other bacterial species than shown here for *Listeria*, including the efficiency of this function, such as high cell yield with remarkably little nutrient requirement from stored reserves [19]. By releasing cells that may find a less hostile environment downstream, such biofilm-to-planktonic cell yield thus also serves as a survival strategy in addition to contributing to proliferation.

The biofilms’ response to treatment showed interesting trends. When cultivated at 10 °C, the biofilms sustained cell release 1 h after exposure to all four PAA concentrations, at even higher numbers at the two lower PAA concentrations (Figure 3). When cultivated at 22 °C, the biofilms sustained cell release 1 h after exposure to the three lower PAA concentrations but not at the highest concentration, whereas release was sustained only at the two lower concentrations at 37 °C (Figure 4 and Figure 5, respectively). However, the trends did not follow the same degree of predictability for subsequent days post-treatment: with complete cessation of cell release lasting two and six days before recovery following treatment with 0.5% and 1.5% PAA, respectively at 10 °C compared to a 2-day cessation only following treatment 1.5% PAA at 22 °C, and a 2-day cessation following treatment with 0.5% and 1.5% PAA, respectively at 37 °C. Biofilm-to-planktonic cell release often peaked after 48 h, but it took notably longer for CO_2_ profiles to stabilize. The latter showed interesting patterns. For example, a typical drop to below CO_2_ detection limits followed treatment, with the duration of minimal metabolic activity (CO_2_ production rate) showing a strong relationship with the different PAA concentrations applied. Absorbance measurements (biofilm biomass) showed some trends, although not as markedly as cell yield or metabolic activity, with much greater fluctuation than previously measured for *Pseudomonas* biofilms [17,20]. As suggested above, the formation and sloughing of ball-shaped cell clusters may be a reason for this. The cell-yield phenomenon was clearly not compromised over the long run, as seen in Figure 3a,c,e. Nevertheless, both biofilm biomass and metabolic activity showed trends that suggest sloughing events, e.g., the red line in Figure 3b at t = approximately 180, 210, 250, and 350 h, a phenomenon earlier reported by Kroukamp et al. [22], which demonstrate the stochastic nature of biofilms. None of these events (rapid decrease in biofilm biomass as measured by optical density) were accompanied by spikes in planktonic cell yield in the effluent, which may be due to the timing of sampling of the effluent.

When compared to a variety of other pure culture and mixed community biofilms from previous studies by the authors, these *Listeria* biofilms had a distinct feature reminiscent of a tumbleweed plant in that it accumulated as clusters, visible to the unaided eye, that periodically moved downstream (Figure 6). It is possible that these clusters consist primarily of a diffuse extracellular matrix with relatively few cells, which may be a survival mechanism where these cells survive in a viable but nonculturable state, with only metabolic maintenance at play. This may be an explanation for the data shown in Figure 3h, where CO_2_ production remained below detection after PAA treatment (deviating from the comparable trends between biofilm biomass and metabolic activity at the lower PAA dosages), accompanied by a steady decline of biofilm biomass and zero release of cultivable cells that would suggest that there were no surviving cells in the biofilms. However, as shown by Bester et al. [25], biofilms accumulate sufficient storage reserves to be able to release 10^5^ to 10^6^ CFU.mL^−1^ planktonic cells during an 8-day starvation period without exogenous carbon supply. Similar to the data in Figure 3h, the starved biofilms in the Bester et al. study produced no measurable CO_2_ during the starvation period. The mobile clusters observed in the present study may have the same function, which may contribute to *Listeria’s* survival success in persisting in food processing and other facilities, where it presents a serious challenge to quality control.

The fact that biofilm-to-planktonic cell release reached maximum values before measured biofilm biomass warrants further investigation. While biofilm biomass control is important where these structures interfere with the flow (loss of hydraulic capacity) and heat exchange and provide protection for microbes involved in biocorrosion, the results shown here suggest that for industries with a risk of product contamination, not only biofilm accumulation is of concern but also their release of cells. This also applies to health care [26]. A plausible explanation for the apparent poor correlation between overall biofilm biomass (thickness and density) and cell release is that the latter occurs primarily at the interface between the biofilm and the bulk aqueous phase, where there is a zone ideal for cell growth, e.g., optimum gas exchange, nutrient supply and removal of metabolites. Microorganisms in this zone could be viewed as surface-associated cells that can either become part of an expanding biofilm matrix or leave this zone/be swept away, with the population size and activity in this zone being influenced by the physical (e.g., flow) and chemical (e.g., nutrients, inhibitors) environment, as well as genotypic switches and variation (e.g., cell motility and biofilm topography). In addition to being the primary area of biofilm cell release, this zone is, therefore, also the primary area of CO_2_ production, and as the first line of contact with the bulk aqueous phase, it will also be most responsive to environmental changes such as exposure to antimicrobials.

The results presented here focus on the treatment of biofilms at the onset of steady state (middle to late stage/mature biofilms) as measured by overall biofilm metabolic activity and biofilm biomass accumulation. Mature biofilms are particularly recalcitrant and resistant, while early-stage biofilms are more vulnerable and susceptible to treatment [27,28]. The treatment of the *Listeria* biofilms during the early stages of development will likely confer greater PAA potency and hinder biofilm maturation. This may manifest in prolonged metabolic suppression, stabilization/reduction in biofilm biomass, and the inhibition/reduction of biofilm-planktonic cell release, similar to the results seen for the higher PAA concentration used in this study. Proactive treatment of early-stage biofilms, rather than mature biofilms, may result in better overall sanitation outcomes. The risk of antimicrobial resistance (AMR) development associated with routine, sub-lethal exposure of biofilms to antimicrobial agents should not be disregarded since biofilms are conventionally viewed as reservoirs of AMR [27,28,29]. The proactive treatment and management of early-stage biofilms informed by better monitoring of biofilm parameters may, therefore, also assist in reducing the risk of AMR development. This is of particular importance in the FPE as it is highly unlikely that these facilities can adopt and maintain a microbe-free environment; a more realistic goal of efficient management of microbial loads may be more attainable.

Overall, the data shows the potential for biofilms to facilitate the dissemination of pathogens and spoilage-causing microbes. Release of ball-shaped microcolonies by *Listeria* may pose an additional challenge to sanitation programs in industries such as FPE, with the thick EPS matrix, as visible in Figure 6, providing physical protection to the cells, in addition to the potential for increased resistance due to nutrient limitation and associated slower growth. Measuring planktonic cell numbers is time-consuming when relying on cultivation or requires specialist infrastructure such as flow cytometry.

## 4. Materials and Methods

### 4.1. Microorganisms and Cultivation Conditions

*Listeria monocytogenes* serovar ½a EGD-e was used for all experimentation and was cultivated aerobically in 10% (*w/v*) Brain Heart Infusion broth (BHI) at various temperatures (10 °C, 22 °C & 37 °C). All precultures were incubated at the relevant temperature for 18 to 24 h, prior to standardization, to an optical density of 0.1 at 595 nm (OD_595nm_) in sterile 10% BHI before experimental use unless otherwise stated. This cell suspension corresponds to approximately 1.0 × 10^9^ CFU.mL^−1^. All reagents were purchased from Sigma-Aldrich (Boksburg, South Africa) unless otherwise stated.

### 4.2. Food-Grade Peracetic Acid (PAA) Sanitizer Formulation and Usage

A commercial food-grade, non-foaming peracetic acid sanitizer (PAA) was obtained from a professional cleaning and sanitation company in South Africa, which predominantly supplies the food and beverage industry. The concentration of the stock PAA solution was ≈100,000 mg/L^−1^ (10% *v/v*) peracetic acid and ≈100,000 mg/L^−1^ (10% *v/v*) hydrogen peroxide (H_2_O_2_). Due to the reactive nature of PAA and limited shelf life (1 year from the date of manufacture), all experimental work was conducted with the same batch of PAA. The PAA and H_2_O_2_ concentrations were determined using colorimetric test strips. PAA concentration was quantified using the Merck MQuant Peracetic Acid Test for 5–50 mg/L^−1^ and 100–500 mg/L^−1,^ and H_2_O_2_ was quantified using the IDS Peroxide 0–100 mg/L^−1^ test strips (RS Components). All working stocks of PAA were freshly diluted in sterile dH_2_O, and the concentration was determined for each compound prior to its same-day use.

### 4.3. Minimum Inhibitory Concentration (MIC) Determination

The standard broth dilution method was used to determine the minimum inhibitory concentration of the PAA sanitizer [30,31]. Briefly, a water reservoir was created in a 96-well microtiter plate by pipetting 250 μL into each well around the perimeter of the plate to minimize the influence of evaporation on test wells. A *L. monocytogenes* Serovar ½a EGD-e preculture was standardized to an OD_595nm_ of 0.2 (double the final OD_595nm_) in fresh, sterile 10% BHI. The working stock of PAA was diluted in 10% BHI at twice the concentration (10% *v/v*) of the desired starting concentration. The range of sanitizer concentrations to be tested was prepared using doubling dilutions (5% *v/v* to 0.01% *v/v* final concentration per well). Thereafter 100 μL of the standardized preculture was added to the PAA and 10% BHI (100 μL) and the control wells (containing only 100 μL BHI), resulting in approx. 1.0 × 10^9^ CFU.mL^−1^ per well (final volume of 200 μL). Uninoculated wells (negative control) contained only PAA and BHI across the same concentration range. The inoculated microtiter plates were incubated for 24 h at 10 °C, 22 °C, and 37 °C, respectively. Microbial growth was quantified spectrophotometrically at 595 nm using a plate reader (Bio-Rad, Hercules, CA, USA). The MIC assays were performed in biological triplicate with three replicate plates.

### 4.4. In-Situ Biofilm Biomass and Metabolic Activity Monitoring under Continuous-Flow Conditions

The silicone tubing in the CEMS-BioSpec system was disinfected with 3.5% (*v/v*) sodium hypochlorite for 2 h and then rinsed with sterile distilled water overnight. Sterile BHI was introduced using a Watson Marlow 205S peristaltic pump. The system was inoculated with 1 mL of a standardized *L. monocytogenes* serovar ½a EGD-e preculture, with the flow ceased for a period of 60 min to allow for colonization of the tubing.

The flow of growth medium (BHI) was initiated at a flow rate of 12.5 mL/h^−1^ with the temperature controlled at 10 °C, 22 °C, or 37 °C using a heating-cooling water bath. The biofilms were cultivated to biomass and metabolic steady state (stabilization of values) at each of the three temperatures prior to the commencement of dosing with pre-determined concentrations of the PAA sanitizer. The dosing period was 20 min, which is twice the duration recommended by the manufacturer. The PAA, freshly diluted in sterile dH_2_O, was introduced into the CEMS-BioSpec system by connecting it to a glass bubble-trap with multiple side-ports located upstream of the peristaltic pump. The inflow of growth medium was thus replaced with the PAA sanitizer dilutions for the duration of the dosing period. The PAA was applied at several concentrations, including the MIC (determined here as 0.16% *v/v*), 0.5% *v/v* (±3× MIC and minimum dose recommended by the manufacturer), 1.5% *v/v* (±10X MIC) and 4.0% *v/v* (maximum recommended dose). Sterile water (0% PAA sanitizer) was included as a control to account for the influence of water only on the biofilms’ response. After dosing, the biofilms were allowed to recover by the re-introduction of the growth medium only for the duration of the experiment.

The concentration of planktonic cells derived from the biofilm was determined at 24 h-intervals for the entire experimental period, 1 h after dosing, and subsequently collected every 24 h until the biofilm had recovered. Briefly, the effluent was collected from the CEMS-BioSpec system, serially diluted in Dey-Engley neutralizing broth (BD, South Africa), and plated on agar-solidified BHI medium. Plates were incubated for 24 h at 22 °C prior to counting the number of colonies. Biofilm cultivation was performed in duplicate, and plate counts were done in triplicate for each PAA concentration.

### 4.5. Abiotic Controls for Continuous-Flow Biofilm Cultivation

The abiotic response of the CEMS-BioSpec system to the BHI only and sterile water with various concentrations of the PAA sanitizer (0%, 0.16%, 0.5%, 1.5%, and 4.0% *v/v* PAA) was assessed. In essence, an identical experimental setup was followed as above with the introduction of sterile BHI into the system. The flow of growth medium, water, and diluted PAA was maintained until a steady state was achieved for both the BioSpec absorbance and CEMS CO_2_ evolution rate measurements, followed by the sequential introduction of BHI and increasing concentrations of PAA. BHI was reintroduced between each concentration of PAA and prior to the termination of the experiment. The abiotic controls were replicated at 10 °C, 22 °C, and 37 °C.

### 4.6. Statistical Analysis

Analysis of variance (ANOVA) with Tukey post-hoc tests were conducted for the microplate datasets utilizing the IBM SPSS 22 software package (*p* < 0.05).

## 5. Conclusions

Ultimately, a combination of techniques should facilitate improved management of water quality and prevention of undesired microbial fouling. For industrial applications, the rapid response of microbial CO_2_ production to chemical treatment should have utility for antimicrobial application, while biofilm biomass measurement shows promise as a direct measurement to monitor biodispersant efficacy. Batch techniques provide valuable information related to almost every field of microbiological research; however, it is evident that the addition of continuous-flow systems can provide valuable additional information, e.g., to delineate form–function relationships in biofilms. In the case of the observations made in this study, it was interesting to note that the production of planktonic cells for proliferation (function) remained remarkably constant regardless of the biofilm biomass (form) and metabolic activity.

## Figures and Tables

**Figure 1 antibiotics-12-00209-f001:**
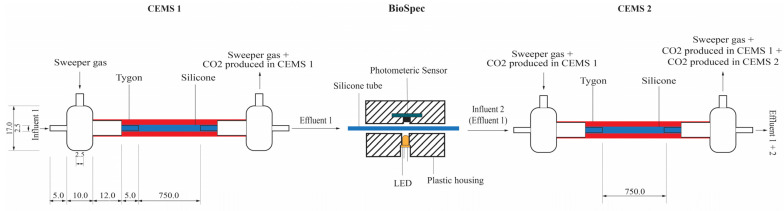
Schematic of the CEMS-BioSpec system used for the real-time monitoring of biofilm parameters under continuous flow conditions. The CO_2_-free sweeper gas is introduced under controlled flow into the annular space of CEMS (red shaded region), allowing for the collection of biofilm-evolved CO_2_ and subsequent analysis by a downstream infrared CO_2_ analyzer. Biofilm biomass was measured between the two CEMS units (BioSpec) via the internal silicone tube (blue shaded region, containing biofilm biomass) being passed through a cavity with a LED illuminating the tube from one side and a digital light sensor on the transverse side measuring the amount of illumination absorbed by biomass in the tube. All dimensions are to the nearest mm [20].

**Figure 2 antibiotics-12-00209-f002:**
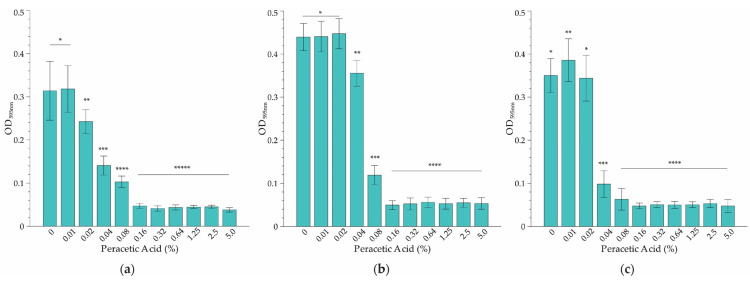
The minimum concentration of PAA sanitizer that inhibits growth (MIC) of *L. monocytogenes* serovar ½a EGD-e as determined at (**a**) 10 °C, (**b**) 22 °C and (**c**) 37 °C using the broth dilution assay. Each bar represents the average of three replicates of three independent experiments, and error bars indicate the standard deviation (*n* = 9). Significance differences between concentrations at each temperature are indicated by *, **, ***, **** and ***** respectively as determined by ANOVA (*p* > 0.05).

**Figure 3 antibiotics-12-00209-f003:**
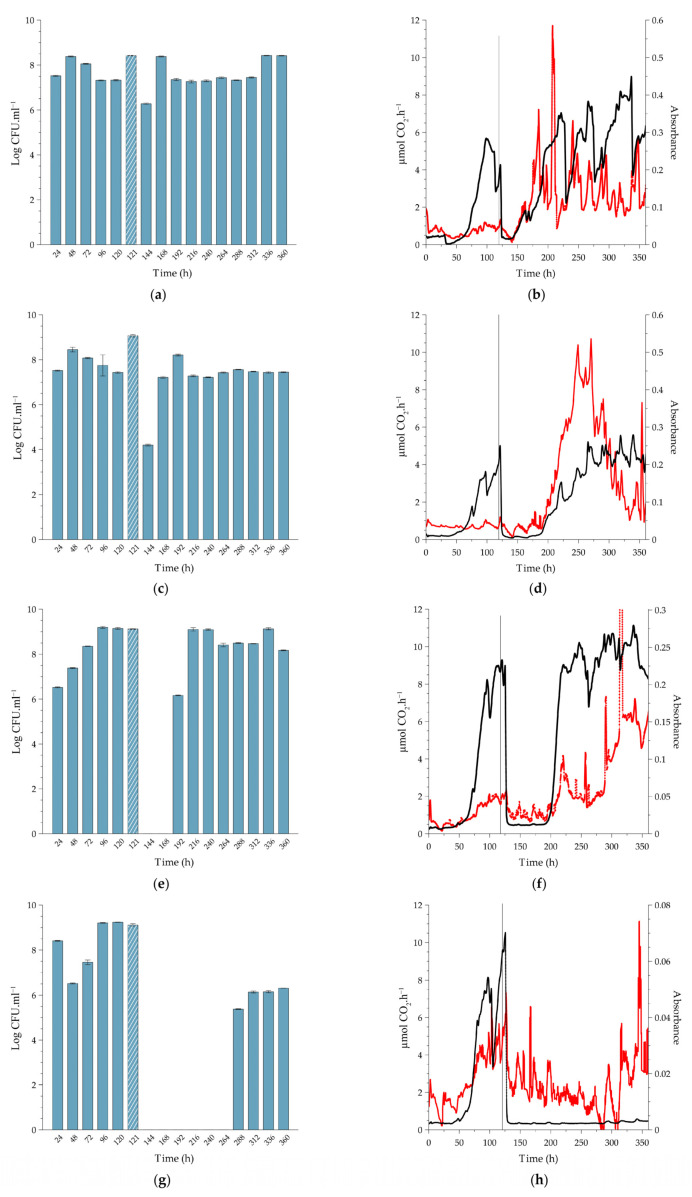
Sessile growth of L. monocytogenes serovar ½a EGDe at 10 °C, exposed to a single (slug) dose of peracetic acid (PAA) for a 20-min period under continuous flow conditions. The biofilms were cultivated in the combined CEMS-BioSpec system on 10% BHI. Planktonic cell yields from sessile communities were determined before and after dosing with increasing concentrations of PAA; (**a**) 0.16% PAA (MIC), (**c**) 0.5% PAA (minimum recommended concentration), (**e**) 1.5% PAA (10-fold increase from MIC) and (**g**) 4.0% PAA (maximum recommended concentration). Crosshatched bars represent planktonic cell yields 1 h after PAA dosing (121 h) of the sessile communities. Each bar represents the logged mean of triple plate counts, with the error bar representing the standard deviation. All data are representative of biological duplicates. Changes in biofilm metabolic activity (black line) and biofilm biomass (red line) were monitored in real-time. The introduction of PAA into the system at 120 h is indicated by a vertical grey line. Biofilms were dosed with increasing concentrations of PAA; (**b**) 0.16% PAA (MIC), (**d**) 0.5% PAA (minimum manufacturer recommended concentration), (**f**) 1.5% PAA (10-fold increase of MIC), and (**h**) 4.0% PAA (maximum manufacturer recommended concentration). Figure 3b,d,f,h are representative data from duplicate biofilms. Each bar in Figure 3a,c,e,g represents the average of triplicate plate counts of duplicate biofilms, and error bars indicate the standard deviation.

**Figure 4 antibiotics-12-00209-f004:**
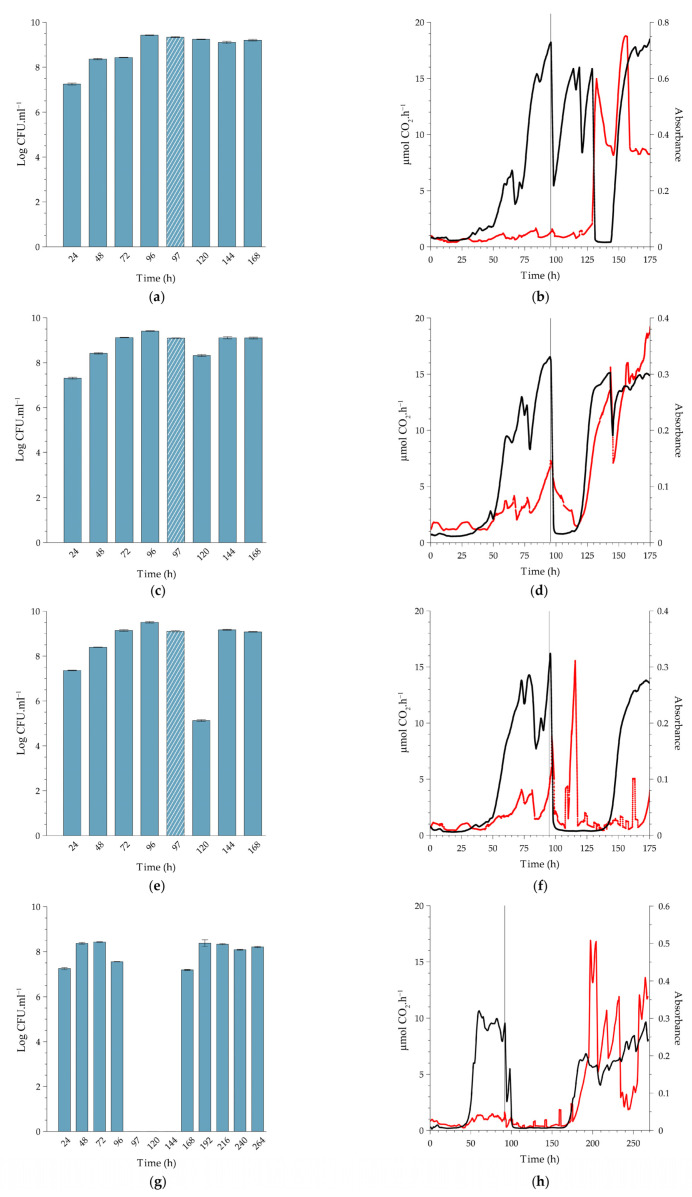
Biofilm growth of *L. monocytogenes* serovar ½a EGD-e at 22 °C, which is representative of room temperature. The biofilms were exposed to a single dose of peracetic acid (PAA) for 20 min under continuous flow conditions. Planktonic cell yields from sessile communities were determined before and after dosing with increasing concentrations of PAA; (**a**) 0.16% PAA (MIC), (**c**) 0.5% PAA (minimum recommended concentration), (**e**) 1.5% PAA (10-fold increase from MIC) and (**g**) 4.0% PAA (maximum recommended concentration). Cross-hatched bars represent planktonic cell yields 1 h after PAA dosing (97 h) of the sessile communities. All data are a representation of biological duplicates. Each bar represents the logged mean of triple plate counts, with the error bar representing the standard deviation. Changes in biofilm metabolic activity (black line) and biofilm biomass (red line) were monitored in real-time, with the dosing period indicated by a vertical black line at 96 h. Biofilms were dosed with increasing concentrations of PAA; (**b**) 0.16% PAA (MIC), (**d**) 0.5% PAA (minimum manufacturer recommended concentration), (**f**) 1.5% PAA (10-fold increase of MIC), and (**h**) 4.0% PAA (maximum manufacturer recommended concentration). Figure 4b,d,f,h are representative data from duplicate biofilms. Each bar in Figure 4a,c,e,g represent the average of triplicate plate counts of duplicate biofilms, and error bars indicate the standard deviation.

**Figure 5 antibiotics-12-00209-f005:**
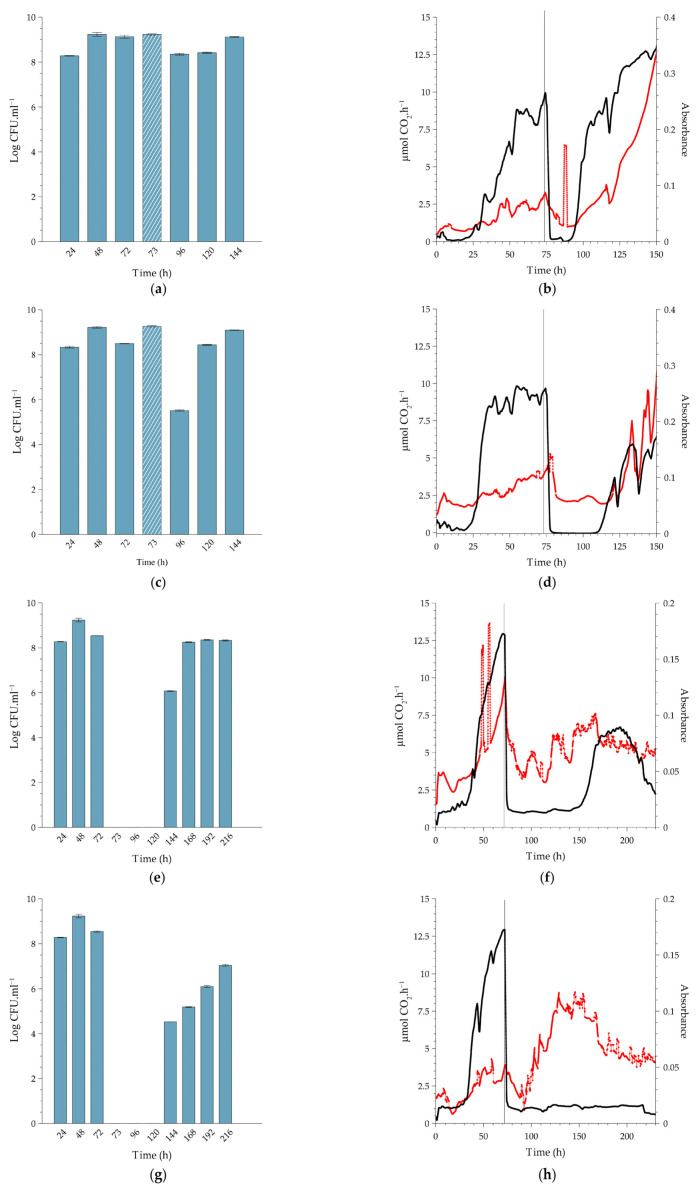
Sessile growth of *L. monocytogenes* serovar ½a EGD-e at 37 °C (body temperature), exposed to a single (slug) dose of peracetic acid (PAA) for a 20-min dosing period under continuous flow conditions. Planktonic cell yields from sessile communities were determined before and after dosing with increasing concentrations of PAA; (**a**) 0.16% PAA (MIC), (**c**) 0.5% PAA (minimum recommended concentration), (**e**) 1.5% PAA (10-fold increase from MIC) and (**g**) 4.0% PAA (maximum recommended concentration). Cross-hatched bars represent planktonic cell yields 1 h after dosing (73 h) of the sessile communities. All data are a representation of biological duplicates. Each bar represents the logged mean of triple plate counts, with the error bar representing the standard deviation. Changes in biofilm metabolic activity (black line) and biofilm biomass (red line) were monitored in real-time with a dosing period indicated by the vertical black line at 72 h. Changes in biofilm metabolic activity (black line) and biofilm biomass (red line) were monitored in real-time, with the dosing period indicated by a vertical black line at 96 h. Biofilms were dosed with increasing concentrations of PAA; (**b**) 0.16% PAA (MIC), (**d**) 0.5% PAA (minimum manufacturer recommended concentration), (**f**) 1.5% PAA (10-fold increase of MIC), and (**h**) 4.0% PAA (maximum manufacturer recommended concentration). Figure 5b,d,f,h are representative data from duplicate biofilms. Each bar in Figure 5a,c,e,g represent the average of triplicate plate counts of duplicate biofilms, and error bars indicate the standard deviation.

**Figure 6 antibiotics-12-00209-f006:**
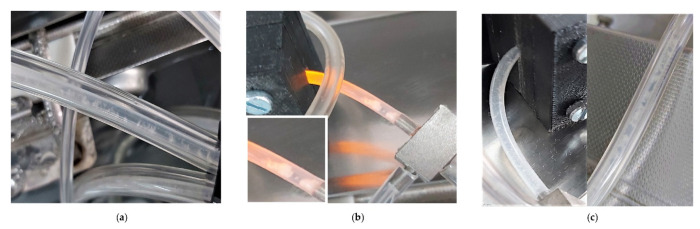
Photographs of sessile growth of *L. monocytogenes* serovar ½a EGD-e at various temperatures in the silicone tubing of the combined CEMS-Biospec system. These Listeria biofilms had a distinct feature reminiscent of tumbleweeds in that they accumulate as clusters that are visible to the unaided eye. These clusters periodically moved downstream through the system. (**a**) The patchy and ball-shaped microcolonies are surrounded by a network of knitted chains near the inflow of the CEMS-Biospec system [24]. (**b**) Larger clusters of the ball-shaped/tumbleweed microcolonies in the tubing just upstream of the BioSpec. (**c**) Smaller clusters of biomass attached to the tubing downstream of the Biospec and CEMS tubing.

## Data Availability

Data is contained within the article or Appendix A.

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
