# Peer review of "Listeria monocytogenes Biofilms Are Planktonic Cell Factories despite Peracetic Acid Exposure under Continuous Flow Conditions"

_antibiotics, 2023, doi:10.3390/antibiotics12020209_

Round 1

Reviewer 1 Report

The paper by Klopper and co-authors is devoted to quantative evaluation of bacterial metabolism and proliferation in L. monocytogenes biofilms treated with increasing concentrations of the disinfectant. The paper is well written and conclusions are well supported by experimental data. There are only a few minor remarks.

1. The description of the CEMS-BioSpec sys-508 used in experiments seems to be moved at the Material and Method section in the last moment. This results in Figure misnumbering, the Figure 1 is the last. Authors can change the Figure order. But to my opinion, it is better to place the description of the experimental system as the first paragraph of Results despite the fact it has been described in the earlier works. The system is not well known, and readers is easier to read it description than to look for other publications.

2. The Fig. 2a is missed.

Reviewer 2 Report

Klopper et al conducted a study investigating the effect of PAA on biofilm formation of Listeria in industrial settings. Although the manuscript is providing some valuable information, there are some concerns which should be addressed.

1. The data analysis part is not adequate. More detailed explanation is needed regarding experimental designs and data analysis. Are all analysis performed using ANOVA?

2. For growth counts, why is standard deviation used instead of standard error? 

3. Figure 1: The use of letters to indicate significance is unclear. Why do the same letters shared between different bars when there are clear differences? Need to use a more standardized format.

4. Metabolic activity, especially with 4% PAA at 4oC. Even though there is a regrowth of bacteria after 144h, why the metabolic activity is not spiking? Are those two experiments done at the same time?

Reviewer 3 Report

Dear authors,

The manuscript meets the scope of the journal and the subject is current and innovative. It's well written, and the data is robust and consistent. The methodology was written in detail and clearly, which assumes that it can be reproduced. In my opinion, all tables, figures and graphs are relevant and contribute to the understanding of the manuscript. The conclusion was clear and well-supported by the data presented. According to the publication, without changes.

Round 2

Reviewer 2 Report

The authors have addressed my initial review comments adequately.